# Prognostic Implications of Blood Immune-Cell Composition in Metastatic Castration-Resistant Prostate Cancer

**DOI:** 10.3390/cancers16142535

**Published:** 2024-07-14

**Authors:** Enrique Perez-Navarro, Vincenza Conteduca, Juan M. Funes, Jose I. Dominguez, Miguel Martin-Serrano, Paolo Cremaschi, Maria Piedad Fernandez-Perez, Teresa Alonso Gordoa, Albert Font, Sergio Vázquez-Estévez, Aránzazu González-del-Alba, Daniel Wetterskog, Begona Mellado, Ovidio Fernandez-Calvo, María José Méndez-Vidal, Miguel Angel Climent, Ignacio Duran, Enrique Gallardo, Angel Rodriguez Sanchez, Carmen Santander, Maria Isabel Sáez, Javier Puente, Julian Tudela, Cecilia Marinas, María Jose López-Andreo, Daniel Castellano, Gerhardt Attard, Enrique Grande, Antonio Rosino, Juan A. Botia, Jose Palma-Mendez, Ugo De Giorgi, Enrique Gonzalez-Billalabeitia

**Affiliations:** 1Department of Medical Oncology, Instituto de Investigación Imas12, Hospital Universitario 12 de Octubre, 28041 Madrid, Spain; enrique.perez2@um.es (E.P.-N.); jmfunes.imas12@h12o.es (J.M.F.); mmartin.imas12@h12o.es (M.M.-S.);; 2Departamento de Ingeniería de la Información y las Comunicaciones, Universidad de Murcia, 30100 Murcia, Spainjtpalma@um.es (J.P.-M.); 3Unit of Medical Oncology and Biomolecular Therapy, Department of Medical and Surgical Sciences, University of Foggia, 71122 Foggia, Italy; 4University College London Cancer Institute, London WC1E 6DD, UK; 5Department of Haematology and Medical Oncology, Hospital Universitario Morales Meseguer, Instituto Murciano de Investigaciones Biosanitarias (IMIB), 30005 Murcia, Spain; 6Medical Oncology Department, Hospital Universitario Ramón y Cajal, 28034 Madrid, Spain; talonso@salud.madrid.org; 7Institut Català dOncologia, Hospital Universitari Germans Trias i Pujol, 08029 Badalona, Spain; 8Hospital Universitario Lucus Augusti, 27003 Lugo, Spain; svazquezestevez@gmail.com; 9Medical Oncology Department, Hospital Universitario Puerta de Hierro-Majadahonda, 28222 Madrid, Spain; 10Medical Oncology Department, Hospital Clínic, 08036 Barcelona, Spain; 11Department of Medical Oncology, Complejo Hospitalario Universitario Ourense, 32005 Orense, Spain; 12Medical Oncology Department, Maimonides Institute for Biomedical Research of Cordoba (IMIBIC), Hospital Universitario Reina Sofía, 14004 Córdoba, Spain; 13Instituto Valenciano de Oncología, 46009 Valencia, Spain; 14Hospital Universitario Marqués de Valdecilla, Instituto de Investigación Valdecilla (IDIVAL), 39011 Santander, Spain; 15Medical Oncology Service, Parc Taulí Hospital Universitari, Institut d’Investigació i Innovació Parc Taulí I3PT, Universitat Autònoma de Barcelona, 08208 Sabadell, Spain; enrqgllrd@gmail.com; 16Department of Medical Oncology, University Hospital of León, 24071 León, Spain; 17Department of Medical Oncology, Hospital Universitario Miguel Servet, 50009 Zaragoza, Spain; 18UGCI Oncología Médica, Hospital Universitario Virgen de la Victoria, 29010 Málaga, Spain; 19Medical Oncology Department, Hospital Clínico San Carlos, Instituto de Investigación Sanitaria del Hospital Clínico San Carlos (IdISSC), CIBERONC, 28040 Madrid, Spain; 20Department of Pathology, Hospital Morales Meseguer, 30008 Murcia, Spain; jtpallares@gmail.com; 21Department of Molecular Biology, Servicio de Apoyo a la Investigación-Instituto Murciano de Investigación Biosanitaria (SAI-IMIB), Universidad de Murcia, 30100 Murcia, Spain; 22Medical Oncology Department, MD Anderson Cancer Center Madrid, Universidad Francisco de Vitoria, 28223 Madrid, Spain; 23Urology Department, Hospital Universitario Morales Meseguer, 30005 Murcia, Spain; 24IRCCS Istituto Romagnolo per lo Studio dei Tumori (IRST) “Dino Amadori”, 47014 Meldola, Italy; 25Facultad de Medicina, Universidad Católica San Antonio de Murcia (UCAM), 30107 Murcia, Spain

**Keywords:** prostate cancer, castration-resistant prostate cancer (CRPC), enzalutamide, whole blood, prognostic factors

## Abstract

**Simple Summary:**

Metastatic castration-resistant prostate cancer (mCRPC) represents a lethal stage of prostate cancer, characterized for its resistance to androgen deprivation therapy and variable survival outcomes. This study investigates how the composition of specific immune cells in the blood affects the prognosis of mCRPC patients who have not yet received chemotherapy. In looking at blood samples taken before treatment with the drug enzalutamide, we discovered significant correlations between lower levels of CD8 T cells and higher levels of monocytes, which were consistently linked to poorer survival rates. The prognostic value of blood CD8 T cells was independently validated in multivariate prognostic models and in an independent cohort of mCRPC patients. This study highlights the feasibility of blood immune-cell profiling in patients included in clinical trials and the association of blood CD8 T cells with the prognosis for mCRPC patients.

**Abstract:**

The prognosis for patients with metastatic castration-resistant prostate cancer (mCRPC) varies, being influenced by blood-related factors such as transcriptional profiling and immune cell ratios. We aimed to address the contribution of distinct whole blood immune cell components to the prognosis of these patients. This study analyzed pre-treatment blood samples from 152 chemotherapy-naive mCRPC patients participating in a phase 2 clinical trial (NCT02288936) and a validation cohort. We used CIBERSORT-X to quantify 22 immune cell types and assessed their prognostic significance using Kaplan–Meier and Cox regression analyses. Reduced CD8 T-cell proportions and elevated monocyte levels were substantially connected with a worse survival. High monocyte counts correlated with a median survival of 32.2 months versus 40.3 months for lower counts (HR: 1.96, 95% CI 1.11–3.45). Low CD8 T-cell levels were associated with a median survival of 31.8 months compared to 40.3 months for higher levels (HR: 1.97, 95% CI 1.11–3.5). These findings were consistent in both the trial and validation cohorts. Multivariate analysis further confirmed the independent prognostic value of CD8 T-cell counts. This study highlights the prognostic implications of specific blood immune cells, suggesting they could serve as biomarkers in mCRPC patient management and should be further explored in clinical trials.

## 1. Introduction

Prostate cancer (PCa) is the most common and the second-most deadly cancer among men [1]. Most patients with metastatic prostate cancer eventually develop castration resistance and succumb to the disease [2,3]. Enzalutamide is an androgen receptor (AR) inhibitor that works by directly binding to AR on its ligand-binding domain and blocking AR activation and signaling, which is critical for the growth and survival of prostate cancer cells. By inhibiting AR signaling, enzalutamide can reduce tumor proliferation and progression. Treatment with enzalutamide has demonstrated to improve survival in advanced prostate cancer patients [4,5,6]. 

The prognosis of metastatic prostate cancer is highly variable, ranging from a few months to several years [7]. Several prognostic factors are available, and distinct prognostic models have been proposed, depending on the clinical scenario and treatment received.

The peripheral blood of patients with mCRPC can be informative of the prognosis of the disease, besides the identification of tumor components. This is due to a complex interplay between the tumor, the bone marrow, and the immune system of the host. Previous reports support the prognostic value of whole-blood RNA signatures [8,9], including genes involved in hematopoiesis and the immune system. In addition, the composition of blood according to the cell types available in the hemogram can also have a prognostic value, as is a high neutrophil-to-lymphocyte ratio [10,11,12,13,14,15,16]. Further understanding of the dynamic interplay between the cancer and the immune cells might provide crucial prognostic information concerning mCRPC.

Blood immune-cell composition includes more than 22 distinct cell populations that can be identified either by flow cytometry or recently by blood cell deconvolution using gene expression arrays. CIBERSORT-X is a machine learning deconvolution algorithm validated for blood immune cell analyses [17,18]. It offers technical advantages for sample analysis at central referral laboratories and provides a method that can be implemented in multicenter clinical trials.

In this study, we aimed to understand the contribution of distinct immune cell types to the prognosis of mCRPC patients. We analyzed prospectively collected pre-treatment blood samples from patients included in a phase 2 multicenter biomarker study investigating the use of enzalutamide as a first-line treatment in mCRPC. Results were further validated in an independent cohort of mCRPC patients.

## 2. Materials and Methods

### 2.1. Study Design and Conduct

The PREMIERE trial represented a multicenter, open-label, single-arm, phase 2 clinical trial (NCT02288936). Its purpose was to explore the use of enzalutamide as a first-line treatment option for metastatic castration-resistant prostate cancer (mCRPC). This study was approved by the Germans Trias i Pujol independent review board (IRB) in Spain (AC-14-112-R). A separate validation cohort was established at the Istituto Scientifico Romagnolo per lo Studio e la Cura dei Tumori (IRST) in Meldola, Italy, with approval from their IRB (REC 2192/2013). The PREMIERE trial constituted a cohort of 98 mCRPC patients, all of whom had not received prior chemotherapy treatments. These participants were recruited across 17 established hospitals within Spain. The selection criteria for patient inclusion incorporated individuals with histologically verified prostate adenocarcinoma, documented metastases, and tumor progression, alongside a serum testosterone level equal to or below 50 ng per deciliter, despite the continuation of androgen-deprivation therapy. In addition, the participants were required to have an Eastern Cooperative Oncology Group (ECOG) rating between 0 and 1 and display either for asymptomatic or for mildly symptomatic conditions (a Brief Pain Inventory Short Form question 3 score less than 4). Detailed outcomes, including primary and secondary findings, have been documented in earlier publications [19,20,21,22].

Additionally, a distinct validation cohort was obtained at the Istituto Scientifico Romagnolo per lo Studio e la Cura dei Tumori (IRST) in Meldola, Italy. This was conducted under the approved protocol REC 2192/2013 and comprised 54 mCRPC patients.

### 2.2. Sample Collection

Serial whole-blood samples from all participants were obtained using PAXgene®RNA Blood RNA tube (PreAnalytiX, Qiagen BD, Valencia, Spain). Samples were stored at −80 °C until processing. Time points for whole blood sample collection included the following: before treatment, on-treatment at 12 weeks, and at tumor progression or end of treatment. For this study, we will focus on analyzing samples taken prior to the initiation of treatment.

### 2.3. RNA Extraction and Microarray Analysis

Whole-blood samples collected in PAXgene RNA tubes were subjected to isolation and purification following the manufacturer’s prescribed protocol. Spectrophotometry and electrophoresis on a microfluidic solution with the NanoDrop 2000 (Thermo Scientific, Newark, DE, USA) and the Bioanalyzer 2100 (Agilent Technologies, Palo Alto, CA, USA), respectively, were utilized for RNA quantification and quality assessment. Only purified RNA samples with an RNA integration number (RIN) of seven or above were chosen for subsequent analyses. Complementary DNA (cDNA) was synthesized from 100 ng of each RNA sample using the WT PLUS Reagent Kit (Thermo Scientific, Newark, DE, USA), following the standard protocol. This cDNA was then amplified, fragmented, and labelled with biotin for hybridization.

For hybridization, the GeneChip™ Human Transcriptome Array HTA 2.0 (902162, Affymetrix, ThermoFisher, Newark, DE, USA) was incubated with 5.2 µg of single-strand DNA (ssDNA) over a period of 16 h. Any non-specific probes were then washed away, and the hybridized array was scanned using the GeneChip Scanner 3000 (Affymetrix, Thermo Scientific, Newark, DE, USA).

### 2.4. Gene Expression Analysis

Gene microarray analyses were derived after quality control, background correction, normalization, logarithmic conversion, and removal of batch effects processing using Transcriptome Analysis Console (TAC) software (version 4.0.1, Thermo, Newark, DE, USA). All samples that had not passed quality control were filtered out, and the resulting data were annotated and analyzed by R packages “affy”, “limma” and “pd.hta.2.0” [23,24,25]. Gene identification relied on the presence of a unique Symbol identifier, with duplicates removed to retain distinct isoforms. The complete dataset includes a comprehensive list of all recognized genes, which is available upon request.

The entire microarray datasets can be accessed through the Gene Expression Omnibus (GSE248619).

### 2.5. CTC and AR-V7 Analysis

The Circulating Tumor Cell (CTC) analyses were performed using the AdnaTest platform (Qiagen, Hilden, Germany), following the manufacturer’s instructions. Custom primers were used for the detection of AR-V7 mRNA. The accuracy of the PCR product was confirmed through Sanger sequencing.

### 2.6. Statistical Analysis

R software (version 4.4.0) was used for statistical analyses [26,27,28]. Survival analyses included Cox proportional hazards regression, log-rank, and the Kapplan–Meier method. Hazard ratios (HRs) were reported as relative risks with corresponding 95% confidence intervals (CIs). A two-sided *p* <0.05 was considered statistically significant.

Survival curves were constructed to visualize the impact of various immune cell populations on patient outcomes. Patients were stratified into ‘high’ and ‘low’ groups, defined by whether their immune cell counts were above or below the median values, respectively. This method of stratification facilitates an unbiased comparison of survival outcomes across the groups.

Multivariable, Cox proportional hazards models were employed to evaluate the association between immune cell proportions and patient survival, adjusting for potential confounders such as the Eastern Cooperative Oncology Group (ECOG) performance status, pattern of spread, prostate-specific antigen (PSA) levels, alkaline phosphatase, lactate dehydrogenase (LDH), pain score, as measured by the Brief Pain Inventory, neutrophil-to-lymphocyte ratio, and corticosteroid usage. Adjustments were made to account for these variables, providing a comprehensive evaluation of their independent contributions to survival outcomes.

## 3. Results

### 3.1. Study Population

This study involved 152 chemotherapy-naïve participants diagnosed with mCRPC, with blood samples obtained before treatment with enzalutamide. The training cohort consisted of 98 patients participating in a phase 2 biomarker clinical trial. All patients had pre-treatment whole-blood samples available for analyses. Gene expression arrays were available from 95 patients. Three patients were excluded due to technical challenges stemming from poor-quality RNA (Figure 1). The patients’ characteristics for the training cohort are described in Appendix A.

### 3.2. Blood Immune-Cell Composition

The distribution of the blood immune cells within the training cohort is depicted in Figure 2 and further detailed in Appendix A.

As expected, neutrophils represented the largest segment of immune cells. They were followed, in decreasing order, by resting NK cells, resting and both naïve and activated memory CD4 T-cells, monocytes, CD8 T-cells, memory B-cells, and resting mast cells. Other immune cells were present in a markedly smaller proportion.

### 3.3. Prognostic Significance of Individual Immune Cell Types

We used Cox-regression analyses to address the contribution of individual immune cell types to overall survival in the training cohort. The results are shown in Table 1. We observed that both an increase in monocytes (*p* < 0.05) and a decrease in CD8 T-Cell lymphocytes (*p* < 0.01) were associated with worse prognosis, after correcting for multiple testing. This prognostic significance remained consistent after stratification based on median values, as depicted in Appendix A. Specifically, patients with high-monocytes (median 32.2 months vs. 40.3 months; HR 1.96, 95%, CI 1.11–3.45) and those with low-CD8 T cells (median 31.8 months vs. 40.3 months; HR 0.51, 95%, CI 0.29–0.9) were associated with worse survival (Figure 3A,B), respectively.

These results were validated in an independent cohort (Appendix A), reproducing the adverse prognosis associated with the presence of high monocyte levels (HR 5.41, 95%, CI 2.60–11.3) and a low CD8 T-cell proportion (HR 0.48, 95% CI 0.25–0.92) (Figure 3C,D), respectively.

We then analyzed the contribution of monocytes and CD8 T-lymphocytes to other well-established prognostic variables, including ECOG, pattern of spread, PSA, alkaline phosphatase, LDH, pain score, neutrophil-to-lymphocyte ratio, and use of corticoids. This multivariable analysis is shown in Table 2.

The CD8 T-cell prognostic value was also independent of known molecular variables such as AR gain or the presence of CTCs. The data are shown in Table 3. The CD8 T cells retained their independent prognostic association with survival, reinforcing their pivotal role as a prognostic factor.

## 4. Discussion

This is the first study to demonstrate that the proportion of blood monocytes and CD8 T cells are prognostic in mCRPC patients. We studied twenty-two blood immune cell types in pre-treatment samples from mCRPC patients included in a multicenter phase 2 biomarker clinical trial using enzalutamide. Our observations revealed that the presence of high monocytes and low CD8 T cells was associated with worse survival. These results were validated in an independent cohort of mCRPC patients. Low CD8 T cells retained independent prognostic significance when included in a validated clinical prognostic model. These results confirm the feasibility of analyzing the immune cell components in a central laboratory in samples obtained from patients participating in a multicenter clinical trial. These results could be valuable for patient stratification in future studies, in particular for clinical trials involving immune cell-activating agents.

The blood of patients with mCRPC has been previously demonstrated to be informative of the prognosis of the disease, besides the presence of tumor components. These studies include gene expression analyses [8,9] and the relative proportion of cells obtained from the hemogram, in particular the neutrophil-to-lymphocyte ratio (NLR) [10,11,12,13,14,15,16]. Gene expression analyses in whole blood have demonstrated prognostic significance in advanced prostate cancer. Interestingly, Olmos et al. [9] developed a nine-gene signature associated with T-cell immune response, and Ross et al. [8] developed a six-gene expression signature, including genes related to the regulation of the cellular immunity and monocyte differentiation. However, this is the first study to evaluate the contribution of twenty-two distinct blood immune cell components that are not routinely provided in the hemogram.

We assessed the blood immune-cell composition using CIBERSORT, a machine learning computational method for characterizing the cell composition of complex tissues obtained from gene expression arrays. It has been demonstrated to accurately quantify the immune-cell constituents of blood samples [17,18,29], and it accurately correlates with flow cytometry techniques to enumerate the phenotypic repertoire of the twenty-two immune-cell subsets included for the analyses [29]. It constitutes a validated method that can substitute flow cytometry for the assessment of immune blood cell components, a technique difficult to implement for central evaluation in multicenter clinical trials.

Several prognostic models are currently available in mCRPC, based on the clinical scenario and treatment of choice. A model was developed for chemotherapy-naïve mCRPC patients treated with enzalutamide based on the clinical data obtained from the PREVAIL clinical trial, a pivotal phase 3 trial for this indication, and the scenario and treatments used in our study. We have recently updated this prognostic model to include molecular variables such as the presence of circulating tumor cells (CTCs) and the amplification of the androgen receptor (*AR*) in circulating plasma DNA [22]. We addressed the independent prognostic value of these cell components relative to the validated clinical models using a multivariable analysis and demonstrated that the CD8 T-cell lymphocyte proportion remained independently prognostic. Intriguingly, the NLR was not prognostic in our series, probably because most patients had a low NLR, associated with the predominance in our study of patients with low tumor burden.

Testosterone is known to act as an immune-suppressor, affecting T-cell function and blocking IFNγ production [30]. Tumors with greater response to anti-androgen therapy are associated with increased proportion of tumor-associated CD8 T cells in the tumor microenvironment [31]. Enzalutamide is an antiandrogen therapy that has demonstrated to increase PD-L1 [32] and tumor immune-cell infiltration. On the contrary, castration-resistant tumors’ progression on enzalutamide is associated with an immunosuppressive microenvironment [33]. In our study, we observed that in patients progressing on androgen-deprivation therapy, the presence of higher levels of CD8 T cells in blood is associated with improved prognosis in a series of patients treated with enzalutamide. This improved prognosis might be related to an increased immune activation that could contribute to the increased prognosis observed.

Several factors, including steroid treatments, could influence the blood immune cell components and could act as cofounding factors on the prognosis. In particular, steroid treatment is associated with increased neutrophil counts and decreased lymphocyte counts [34]. We noted that a small subgroup of our patients were receiving low-dose steroids at baseline, including six patients receiving low-dose prednisone (≤10 mg). In order to exclude a confounding effect of steroids on the prognostic value of these immune-cell components, we included the use of steroids in a comprehensive multivariable analysis comprising the accepted clinical and laboratory variables, in addition to the prognostic immune cell components. The use of steroids did not modify the prognostic value for CD8 T cells nor for monocytes, and steroids lacked prognostic significance. Therefore, we can conclude that the minority of patients receiving low-dose steroids did not affect the results reported on the prognostic value of these immune cell components (Appendix A).

Immune-cell composition assessment using CIBERSORT, similar to all gene expression-based methods, can be limited by the fidelity of the reference profiles, which can be deviated under particular conditions. In addition, some cell types can be systematically over- or under-estimated, an effect that can be limited by inter-group relative comparisons. However, it is a validated method for the assessment and monitoring of the immune-cell population in complex tissues such as the blood. Another limitation of this study is related to the inclusion of patients with asymptomatic or mildly symptomatic chemo-naïve mCRPC. Despite the fact that this is the most frequent scenario in the clinic, it would also be interesting to address the immune components in the blood of patients with other clinical characteristics, including patients with pain and with aggressive or neuroendocrine features of the disease, and on treatment with other approved agents in this scenario, including other androgen receptor pathway inhibitors (ARPIs), docetaxel, Radium-223, or 177Lu-PSMA.

Our study is unable to address the predictive value of CD8 T cells and monocytes in mCRPC. This question could be assessed in patients participating in a randomized trial preferentially including immunotherapy. A fine dynamic assessment of the blood immune components during treatment could be key to understanding the systemic effects of the prostate cancer treatments, including hormonal agents and the immune-checkpoint inhibitors, among others. This understating is going to be essential to move forward immunotherapy in prostate cancer.

## 5. Conclusions

In conclusion, this study highlights the contribution of blood CD8 T cells and monocytes to the prognosis of patients with mCRPC treated with new hormonal agents. Further understanding of the complex interplay between the immune system, prostate cancer, and the treatments is necessary to design more effective immunotherapies.

## Figures and Tables

**Figure 1 cancers-16-02535-f001:**
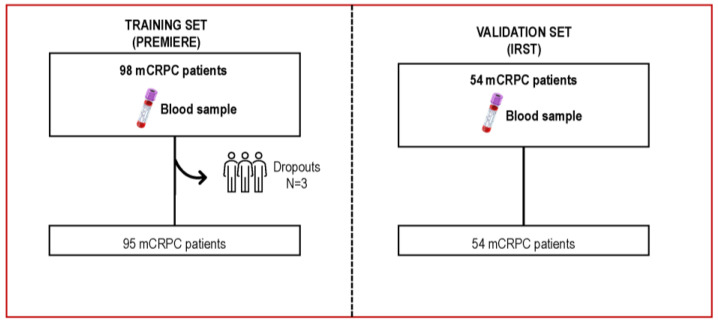
Consort diagram. This study analyzes pre-treatment whole blood samples from mCRPC patients prospectively treated with enzalutamide, including a training cohort comprised of 98 patients from a phase 2 biomarkers clinical trial, and an independent validation cohort including 54 patients.

**Figure 2 cancers-16-02535-f002:**
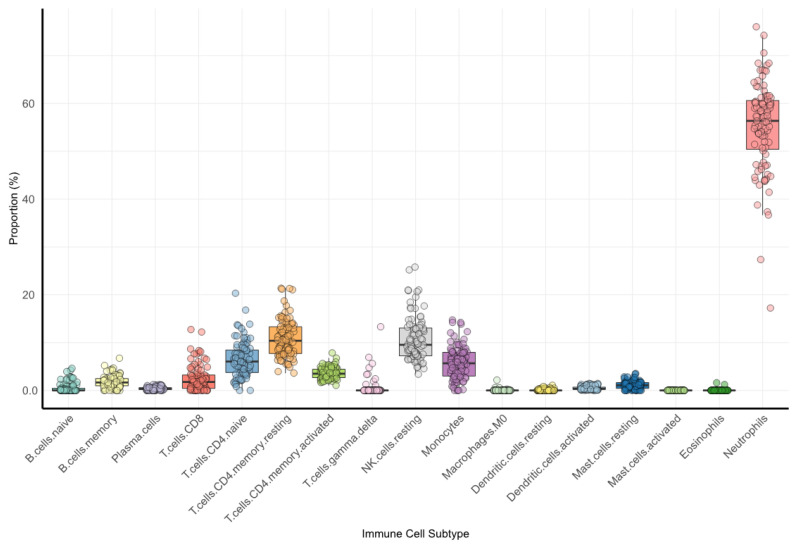
Blood immune-cell composition. The figure shows the relative proportion of immune cell components in the blood in the training set.

**Figure 3 cancers-16-02535-f003:**
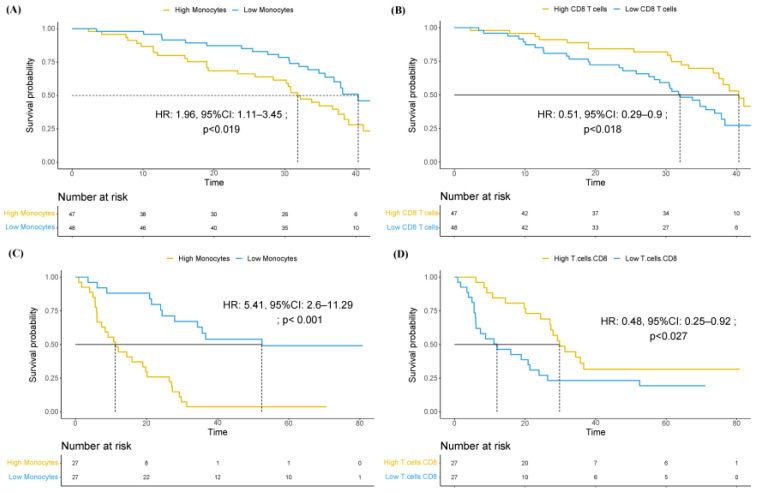
Overall survival analysis in PREMIERE and IRST validation cohorts. Kaplan–Meier survival curves and Cox proportional hazards regression analyses showing the impact of immune cell levels on patient survival in the PREMIERE (**A**,**B**) and IRST validation (**C**,**D**) cohorts. (**A**) Survival based on monocyte counts in PREMIERE cohort. (**B**) Survival based on CD8 T-cell levels in PREMIERE cohort. (**C**) Survival based on monocyte counts in IRST validation cohort. (**D**) Survival based on CD8 T-cell levels in IRST validation cohort. Time is expressed in months. Curves show survival probabilities for groups with varying immune cell levels. Numbers below each graph indicate patients at risk at specific time points. Hazard ratios (HR), 95% confidence intervals (CI), and *p*-values from Cox regression analyses are provided for each comparison.

**Table 1 cancers-16-02535-t001:** Survival analyses by blood immune cell type. Cox-regression survival analyses was conducted for each individual immune cell population.

Blood Immune Cell Type	HR (95% CI)	*p* Value
Memory B cells	1.21 (0.69–2.12)	0.506
Plasma cells	1.32 (0.76–2.31)	0.323
T cells CD8	0.51 (0.29–0.9)	0.018
T cells CD4-naive	1.11 (0.64–1.94)	0.71
T cells CD4 memory, resting	0.72 (0.41–1.26)	0.252
T cells CD4 memory, activated	0.87 (0.50–1.51)	0.617
NK cells, resting	0.92 (0.53–1.61)	0.78
Monocytes	1.96 (1.11–3.45)	0.019
Dendritic cells, activated	1.65 (0.94–2.89)	0.081
Mast cells, resting	0.92 (0.53–1.61)	0.775
Neutrophils	0.98 (0.56–1.71)	0.935

Abbreviations: HR = hazard ratio. CI = confidence interval. HR compares high versus low values, according to the median. *p* values < 0.05 are considered statistically significant. The following cell types were below the quantifiable limit of detection, and survival analyses are not shown: follicular helper T cells, regulatory T cells (Tregs), gamma delta T cells, activated NK cells, naive B cells, macrophages M0, macrophages M1, macrophages M2, resting dendritic cells, activated mast cells, and eosinophils.

**Table 2 cancers-16-02535-t002:** Multivariable analysis including clinical and molecular variables.

Prognostic	HR (95% CI)	*p* Value
ALP_Mod	1.84 (0.99–3.342)	0.055
LDH_Mod	1.91 (1.02–3.60)	0.045
Pattern Of Spread	1.08 (0.24–4.82)	0.922
NLR	0.70 (0.36–1.34)	0.279
BPI	0.59 (0.30–1.14)	0.117
LogPSA	1.44 (1.18–1.76)	0.001
Monocytes	1.05 (0.74–1.49)	0.770
CD8 T cells	0.63 (0.41–0.98)	0.04
ECOG	1.33 (0.71–2.50)	0.371

Abbreviations: ALP = alkaline phosphatase; LDH = lactate dehydrogenase; NLR = neutrophil-to-lymphocyte ratio; ECOG = Eastern Cooperative Oncology Group; BPI: Brief Pain Inventory; CI = confidence interval; HR = hazard ratio. *p* value was calculated using Cox regression.

**Table 3 cancers-16-02535-t003:** Multivariable analysis including ARgain and CTCs.

	HR (95% CI)	*p* Value
T cells CD8	0.54 (0.35–0.83)	0.006
ARgain	6.17 (2.83–13.46)	<0.001
CTCs	4.63 (2.58–8.31)	<0.001

Abbreviations: AR = androgen receptor; CTC = circulating tumor cells.

## Data Availability

The datasets generated and/or analyzed during the current study include gene expression data that have been deposited in the Gene Expression Omnibus (GEO) under the accession number GSE248619. For additional data not publicly available, access can be provided by the corresponding author, EGB, on reasonable request.

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
