# Peer review of "Prognostic Implications of Blood Immune-Cell Composition in Metastatic Castration-Resistant Prostate Cancer"

_cancers, 2024, doi:10.3390/cancers16142535_

Round 1

Reviewer 1 Report

Comments and Suggestions for Authors

Metastatic castration-resistant prostate cancer is lethal disease that needs further therapeutic improvements. In the manuscript «Prognostic implications of blood immune cell composition in 2 Metastatic Castration-Resistant Prostate Cancer» Enrique Perez-Navarro et al investigate the composition of immune cell populations by using gene expression arrays together with CIBERSORT-Xa machine learning deconvolution algorithm, a method that have a great potential for prognosis. 

The topic of the study is interesting. The manuscript is well organized and written.

I raise the following points:

1.     Please define the cut offs. According to what patients were divided into low and high in Figure 3? 

2.     Why healthy controls were not used in this study? 

3.     Authors in lines 221-224 and in Table 3 mention and count CTCs. CTCs are not described in the Materials and Methods. How you define CTCs? What method was used? According to what markers CTCs were detected?

4.     Authors should briefly discuss the mechanism of action of enzalutamide and explain why immune cell populations such as CD8 T-cells and MDSCs could play a role in survival upon this specific treatment. 

Author Response

Comments 1: Please define the cut offs. According to what patients were divided into low and high in Figure 3?

Thank you for your query regarding the cut-offs used to divide patients into high and low groups in Figure 3. The cut-offs for categorizing patients into low and high groups were based on the median values of the respective immune cell populations. Specifically, patients with values below the median were categorized as "low," and those with values above the median were categorized as "high." This method ensures an unbiased and statistically balanced separation of patient groups for survival analysis. 

This clarification has been included in the new version in Material and Methods section (Line 172).

Comments 2:
Why healthy controls were not used in this study?

The primary aim of our research was to investigate the prognostic implications of blood immune cell composition in patients with metastatic castration-resistant prostate cancer (mCRPC) undergoing treatment with enzalutamide. Therefore, our focus was on correlating immune cell populations with clinical outcomes within this specific patient population. Including healthy controls could provide additional insights into the baseline immune cell composition; however, our study was specifically designed to address prognostic factors in a clinical setting. Future studies may consider including healthy controls for a more comprehensive understanding of the immune landscape in mCRPC patients.

Comments 3: Authors in lines 221-224 and in Table 3 mention and count CTCs. CTCs are not described in the Materials and Methods. How you define CTCs? What method was used? According to what markers CTCs were detected?

We appreciate this comment. We have added the following paragraph in methods (Line 161)

Comments 3: Authors should briefly discuss the mechanism of action of enzalutamide and explain why immune cell populations such as CD8 T-cells and MDSCs could play a role in survival upon this specific treatment.

We greatly appreciate this comment by the reviewer. We have added two paragraphs in the manuscript:

(Line 76) Enzalutamide is an androgen receptor (AR) inhibitor that works by directly binding to AR on its ligand binding domain and blocking AR activation and signaling, which is critical for the growth and survival of prostate cancer cells. By inhibiting AR signaling, enzalutamide can reduce tumor proliferation and progression. Treatment with enzalutamide has demonstrated to improve survival in advanced prostate cancer patients [4–6].

(Line 299). Testosterone is known to act as an immune-suppressor, affecting T-cell function and blocking IFNγ production [30]. Tumors with greater response to anti-androgen therapy are associated with increased proportion of tumor-associated CD8 T cells in the tumor microenvironment [31].Enzalutamide is an antiandrogen therapy that has demonstrated to increase PD-L1 [32]and to increase tumor immune-cell infiltration, and on the contrary castration resistant tumors in progression to enzalutamide are associated with an immunosuppressive microenvironment [33]. In our study we observe that in patients in progression to androgen deprivation therapy the presence of higher levels of CD8-T cells in blood is associated with improved prognosis in a series of patients treated with enzalutamide. This improved prognosis might be related with an increased immune activation that could contribute to the increased prognosis observed

Reviewer 2 Report

Comments and Suggestions for Authors

This multi-centre paper makes a seemingly interesting observation regarding prostate cancer prognosis in relation to CD8+ T cells and monocytes in the blood. It is a pity the authors did not speculate on any mechanisms of how the pathogenesis might be affected by these changes in blood WBC composition. A few minor issues:

1.    L56: should be ‘patients.

2.    L65: H not h

3.    L88: ‘as it is a high neutrophile’ should be ‘as is a high neutrophil..,

4.    Fig. 2: some ambiguity in labelling y axis here, neutrophils make up 40-60% of WBCs, the graph shows 0.6%!!!

5.    Fig. 3 legend should read ‘in the Training (A,B) and Validation (C,D)….’

6.    L218: ‘neutrophile’ should be ‘neutrophil’

7.    L238: ‘have previously’ should be ‘has been previously…’

8.    L251: ‘has demonstrated..’ should be ‘has been demonstrated..’

9.    L267: ‘most patient’ should be ‘most patients..’

10. No corresponding author and no academic email

Comments on the Quality of English Language

Minor errors spotted, see report

Author Response

Comments 1: This multi-centre paper makes a seemingly interesting observation regarding prostate cancer prognosis in relation to CD8+ T cells and monocytes in the blood. It is a pity the authors did not speculate on any mechanisms of how the pathogenesis might be affected by these changes in blood WBC composition.

We greatly appreciate this comment by the reviewer. We have included a paragraph in the discussion on this topic:

(Line 299). Testosterone is known to act as an immune-suppressor, affecting T-cell function and blocking IFNγ production [30]. Tumors with greater response to anti-androgen therapy are associated with increased proportion of tumor-associated CD8 T cells in the tumor microenvironment [31].Enzalutamide is an antiandrogen therapy that has demonstrated to increase PD-L1 [32]and to increase tumor immune-cell infiltration, and on the contrary castration resistant tumors in progression to enzalutamide are associated with an immunosuppressive microenvironment [33]. In our study we observe that in patients in progression to androgen deprivation therapy the presence of higher levels of CD8-T cells in blood is associated with improved prognosis in a series of patients treated with enzalutamide. This improved prognosis might be related with an increased immune activation that could contribute to the increased prognosis observed.

Comments 2: A few minor issues.

We sincerely appreciate your detailed and constructive comments. We are pleased to inform you that we have addressed all the minor issues mentioned.

Round 2

Reviewer 1 Report

Comments and Suggestions for Authors

The manuscript is now improved.